METHODS

# A flexible empirical Bayes approach to multivariate multiple regression, and its improved accuracy in predicting multi-tissue gene expression from genotypes

**Fabio Morgante**[1,2,3]*, **Peter Carbonetto**[4,5], **Gao Wang**[4,6,7], **Yuxin Zou**[8,9],
**Abhishek Sarkar**[4¤], **Matthew Stephens**[4,8]*

**1** Center for Human Genetics, Clemson University, Greenwood, South Carolina, United States of America,
**2** Department of Genetics and Biochemistry, Clemson University, Clemson, South Carolina, United States of America, **3** Section of Genetic Medicine, Department of Medicine, University of Chicago, Chicago, Illinois, United States of America, **4** Department of Human Genetics, University of Chicago, Chicago, Illinois, United States of America, **5** Research Computing Center, University of Chicago, Chicago, Illinois, United States of America, **6** Department of Neurology, Columbia University, New York, New York, United States of America, **7** Gertrude H. Sergievsky Center, Columbia University, New York, New York, United States of America, **8** Department of Statistics, University of Chicago, Chicago, Illinois, United States of America, **9** Regeneron Genetics Center, Regeneron Pharmaceuticals Inc., Tarrytown, New York, United States of America

¤ Current address: Vesalius Therapeutics, Inc., Cambridge, Massachusetts, United States of America
* fabiom@clemson.edu (FM); mstephens@uchicago.edu (MS)

**Data Availability Statement:** The genotype and expression data used in our analyses are available from dbGaP (https://www.ncbi.nlm.nih.gov/

## Abstract

Predicting phenotypes from genotypes is a fundamental task in quantitative genetics. With technological advances, it is now possible to measure multiple phenotypes in large samples. Multiple phenotypes can share their genetic component; therefore, modeling these phenotypes jointly may improve prediction accuracy by leveraging *effects that are shared across phenotypes*. However, effects can be shared across phenotypes in a variety of ways, so computationally efficient statistical methods are needed that can accurately and flexibly capture patterns of effect sharing. Here, we describe new Bayesian multivariate, multiple regression methods that, by using flexible priors, are able to model and adapt to different patterns of effect sharing and specificity across phenotypes. Simulation results show that these new methods are fast and improve prediction accuracy compared with existing methods in a wide range of settings where effects are shared. Further, in settings where effects are not shared, our methods still perform competitively with state-of-the-art methods. In real data analyses of expression data in the Genotype Tissue Expression (GTEx) project, our methods improve prediction performance on average for all tissues, with the greatest gains in tissues where effects are strongly shared, and in the tissues with smaller sample sizes. While we use gene expression prediction to illustrate our methods, the methods are generally applicable to any multi-phenotype applications, including prediction of polygenic scores and breeding values. Thus, our methods have the potential to provide improvements across fields and organisms.

projects/gap/cgi-bin/study.cgi?study_id=
phs000424.v8.p2). All code implementing the
simulations, and the compiled results generated
from our simulations have been deposited on
Zenodo (https://doi.org/10.5281/zenodo.8014360).
The methods are implemented in the R package
mr.mash.alpha, available for download at https://
github.com/stephenslab/mr.mash.alpha.

**Funding:** Research reported in this publication was
supported by the National Institute of General
Medical Sciences of the National Institutes of
Health under Award Numbers P20GM139769 and
R35GM146868 to FM. MS acknowledges support
from National Human Genome Research Institute
grant R01HG002585. GW acknowledges support
from National Institute of Aging grant
R01AG076901. The content is solely the
responsibility of the authors and does not
necessarily represent the official views of the
National Institutes of Health. The funders had no
role in study design, data collection and analysis,
decision to publish, or preparation of the
manuscript.

**Competing interests:** The authors have declared
that no competing interests exist.

## Author summary

Predicting phenotypes from genotypes is a fundamental problem in quantitative genetics.
Thanks to recent advances, it is increasingly feasible to collect data on many phenotypes
and genome-wide genotypes in large samples. Here, we tackle the problem of predicting
*multiple phenotypes* from genotypes using a new method based on a multivariate, multiple
linear regression model. Although the use of a multivariate, multiple linear regression
model is not new, in this paper we introduce a flexible and computationally efficient
*empirical Bayes* approach based on this model. This approach uses a prior that captures
how the effects of genotypes on phenotypes are shared across the different phenotypes,
and then the prior is adapted to the data in order to capture the most prominent sharing
patterns present in the data. We assess the benefits of this flexible Bayesian approach in
simulated genetic data sets, and we illustrate its application in predicting gene expression
measured in multiple human tissues. We show that our methods can outperform compet-
ing methods in terms of prediction accuracy, and the computations involved in fitting the
model and making the predictions scale well to large data sets.

## Introduction

Multiple regression has been an important tool in genetics for different tasks relating geno-
types and phenotypes, including discovery, inference, and prediction. For discovery, multiple
regression has been used to fine-map genetic variants discovered by Genome-Wide Associa-
tion Study (GWAS) [1, 2]. For inference, multiple regression has been used to estimate the
proportion of phenotypic variance explained by genetic variants—*i.e.*, "genomic heritability"
or "SNP heritability" [3–5]. For prediction, multiple regression has been used extensively to
predict yet-to-be-observed phenotypes from genotypes. This task is relevant to the prediction
of breeding values for selection purposes in agriculture [6, 7], the prediction of "polygenic
scores" for disease risk and medically relevant phenotypes in human genetics [8–10], and the
prediction of gene expression as an intermediate step in transcriptome-wide association stud-
ies (TWAS) [11, 12]. Traditionally, frequentist multiple regression methods such as penalized
regression and linear mixed models [13–16] have been used for these tasks. However, Bayesian
methods have received particular attention in genetic applications because they provide a nat-
ural way to incorporate prior information about and cope with different genetic architectures.
This attractive feature has spurred the development and application of many Bayesian meth-
ods that differ in their prior distribution on the effect sizes and their approach to computing
posterior distributions [6, 10, 17–27].

   Most multiple regression methods in widespread use are "univariate" in that they model
only one outcome (phenotype). However, many studies involve multiple outcomes that may
share genetic effects [28]. Examples of this include organism-level phenotypes measured in
multiple environments or populations, such as those available in UK Biobank [29] or BioBank
Japan [30], and multiple molecular phenotypes such as the expression levels of multiple genes
in multiple tissues available in reference data sets such as the Genotype Tissue Expression
(GTEx) project [31]. In such cases, joint ("multivariate") modeling of multiple phenotypes can
improve performance over separate univariate analyses that consider one phenotype at a time.
Indeed, multivariate analysis can improve performance even when phenotypes are not geneti-
cally correlated provided that phenotypes are phenotypically correlated [32]. Multivariate anal-
ysis of multiple phenotypes has been shown to improve power to discover associations [33–36]
and accuracy of phenotype prediction [37–40].

However, currently available multivariate multiple regression methods have important limitations. The multivariate versions of popular penalized regression methods (e.g., ridge regression, the Elastic Net, the Lasso implemented in the popular R package `glmnet` [41]) do not allow for missing phenotype values and, more importantly, do not exploit patterns of effect sharing. Urbut *et al* [35] showed the benefits of multivariate methods that learn effect sharing from the data. Multivariate linear mixed models (MLMM) [42] can also learn effect sharing from the data, but they lack flexibility—these models make the "infinitesimal architecture" assumption that every variant has an effect on all phenotypes which is not appropriate for phenotypes with sparse architectures [43]. Bayesian methods are a natural way to achieve flexibility in terms of sparsity of the signal and can learn patterns of effect sharing from the data. These methods include multivariate versions of of the "Bayesian alphabet" methods such as BayesB, BayesCΠ, and the Bayesian Lasso [44, 45]. However, despite the added flexibility compared to the MLMM model, the prior families used in existing multivariate Bayesian methods make them relatively inflexible to cope with the complex distribution of effect sizes that many complex traits have. In fact, most of those methods either have a single distribution or a "spike-and-slab" type of prior, with only one non-point-mass ("slab") component. In addition, the use of computationally intensive Markov Chain Monte Carlo (MCMC) algorithms for model fitting makes the multivariate Bayesian alphabet methods impractical in many "genome-wide" settings, even with a moderate number of phenotypes.

To overcome these limitations, we introduce a new method, "Multiple Regression with Multivariate Adaptive Shrinkage" or "*mr.mash*". *mr.mash* is a Bayesian multivariate, multiple regression method that is able to learn complex patterns of effect sharing from the data while also being computationally efficient. We achieve this by combining three powerful ideas: (1) flexible prior distributions that allow for complex patterns of effect sharing across phenotypes; (2) empirical Bayes for adapting the priors to the data; and (3) variational inference for fast Bayesian computations. In particular, this work integrates previous work by Urbut *et al* [35] (ideas 1 and 2) with previous work by Carbonetto and Stephens [20] (idea 3) into a single framework, and extends the methods of Kim *et al* [27] to the multivariate setting. We show via extensive simulations of multi-tissue gene expression prediction from genotypes that *mr.mash* can adapt to complex patterns of effect sharing and specificity, and outperforms competing methods. These results are confirmed in analyses of real data from the Genotype Tissue Expression (GTEx) project [31], demonstrating the potential for our method to more accurately impute expression levels, as is required for TWAS [11, 12]. Although this work was primarily motivated by our interest in improving predictions of gene expression, *mr.mash* can be applied to other settings where predictions from multivariate multiple regression are desired, such as computing polygenic scores or breeding values.

## Description of the method

We consider the multivariate multiple regression model of outcomes $Y$ on predictors $X$,

$$
\begin{aligned}
Y &= XB + E \\
E &\sim MN_{n \times r}(\mathbf{0}, I_n, V),
\end{aligned}
\tag{1}
$$

where $Y$ is an $n \times r$ matrix of $r$ outcomes observed in $n$ samples (possibly containing missing values), $X$ is an $n \times p$ matrix of $p$ predictors observed in the same $n$ samples, $B$ is the $p \times r$ matrix of effects, $E$ is an $n \times r$ matrix of residuals, $I_n$ is the $n \times n$ identity matrix, and $MN_{n \times r}(M, U, V)$ is the matrix normal distribution with mean $M \in \mathbb{R}^{n \times r}$ and covariance matrices $U \in \mathbb{R}^{n \times n}$, $V \in \mathbb{R}^{r \times r}$ [46, 47]. For example, in our application later we aim to predict gene expression in multiple tissues from genetic variant genotypes, so $y_{is}$ is the observed gene

expression in individual $i$ and tissue $s$, and $x_{ij}$ is the genotype of individual $i$ at genetic variant $j$. (In practice, an intercept $\boldsymbol{b}_0 \in \mathbb{R}^r$ is included in the regression model, but we leave this detail out here; full details of the model are given in the S1 Text.)

Let $\boldsymbol{b}_j$ denote the $j$th row of $\boldsymbol{B}$ (as a column vector); thus, $\boldsymbol{b}_j$ is an $r$-vector reflecting the effects of variable $j$ on the $r$ outcomes. To capture the potential similarity of the effects among the different outcomes, we use a mixture of multivariate normals prior on $\boldsymbol{b}_j$ [35],

$$\boldsymbol{b}_j \mid \boldsymbol{w}_0, \mathscr{S}_0 \sim \sum_{k=1}^{K} w_{0,k} N_r(\boldsymbol{0}, \boldsymbol{S}_{0,k}), \tag{2}$$

where $N_r(\boldsymbol{\mu}, \boldsymbol{\Sigma})$ denotes the multivariate normal distribution on $\mathbb{R}^r$ with mean $\boldsymbol{\mu}$ and covariance $\boldsymbol{\Sigma}$, $\boldsymbol{w}_0 := (w_{0,1}, \ldots, w_{0,K})$ is a set of mixture weights (non-negative and summing to one), and $\mathscr{S}_0 := \{\boldsymbol{S}_{0,1}, \ldots, \boldsymbol{S}_{0,K}\}$ denotes a collection of $r \times r$ covariance matrices. Following [35], we assume that the covariance matrices $\mathscr{S}_0$ are pre-specified, and treat the mixture weights $\boldsymbol{w}_0$ as parameters to be estimated from the data. The idea is that the collection of matrices $\mathscr{S}_0$ should be chosen to include a wide variety of potential effect sharing patterns; the estimated $\boldsymbol{w}_0$ should then assign most weight to the sharing patterns that are present in the data and little or no weight to patterns that are inconsistent with the data. We discuss selection of suitable covariance matrices $\mathscr{S}_0$ in S1 Text.

Since our approach combines the multiple regression model (1) with multivariate adaptive shrinkage priors (2) from [35], we call our approach "*mr.mash*", which is short for "*Multiple Regression* with *Multivariate Adaptive Shrinkage*".

## Variational empirical Bayes for *mr.mash*

To fit the *mr.mash* model we use variational inference methods [48, 49] which have been successfully applied to fit univariate multiple regressions [10, 20, 24, 25, 27, 50]. Variational inference recasts the posterior computation as an optimization problem. Specifically, we seek a distribution $q(\boldsymbol{B})$ which approximates the true posterior distribution, $p(\boldsymbol{B} \mid \boldsymbol{X}, \boldsymbol{Y}, \boldsymbol{V}, \boldsymbol{w}_0, \mathscr{S}_0)$. By imposing simple conditional independence assumptions on the approximate posterior distribution, $q(\boldsymbol{B})$, the posterior computations and optimization of $q(\boldsymbol{B})$ become tractable.

In addition to approximating the posterior distribution of $\boldsymbol{B}$, the variational approach also provides a way to estimate the model parameters, $\boldsymbol{w}_0$ and $\boldsymbol{V}$, by maximizing an approximation to the marginal likelihood, $p(\boldsymbol{Y} \mid \boldsymbol{X}, \boldsymbol{V}, \boldsymbol{w}_0, \mathscr{S}_0)$, which is known as the "evidence lower bound" (ELBO) [48]. This approach was called "variational empirical Bayes" in [51], although this idea of fitting the model parameters by maximizing an approximate marginal likelihood dates back to earlier work [52, 53].

The variational empirical Bayes algorithm for *mr.mash* is outlined in Algorithm 1 of the S1 Text. (This algorithm also handles imputation of missing data which we explain in the next section.) This algorithm has an inner loop over the variables (the genetic variants) $j = 1, \ldots, p$, which can be viewed as a coordinate ascent algorithm for fitting the approximate *mr.mash* posterior, $q(\boldsymbol{B})$, under the assumption that the $\boldsymbol{b}_j$'s are conditionally independent *a posteriori* (S1 Text).

The core of the algorithm's inner loop is the "BMSR-mix" step. This computes the posterior distribution of a *mr.mash* model containing just a single variable. ("BMSR-mix" is short for "Bayesian multivariate simple regression with a mixture prior.") The posterior distribution of $\boldsymbol{b}_j$ is a mixture of multivariate normals (S1 Text), so the posterior distribution is therefore fully specified by the posterior mixture weights $w_{1,k}$, the posterior means $\boldsymbol{b}_{1,k}$, and the posterior covariances $\boldsymbol{S}_{1,k}$. The underlying BMSR-mix computations have closed-form expressions. However, the computations can be expensive, particularly when $r$ and/or $K$ are large, so this step represents the main computational bottleneck of *mr.mash*.

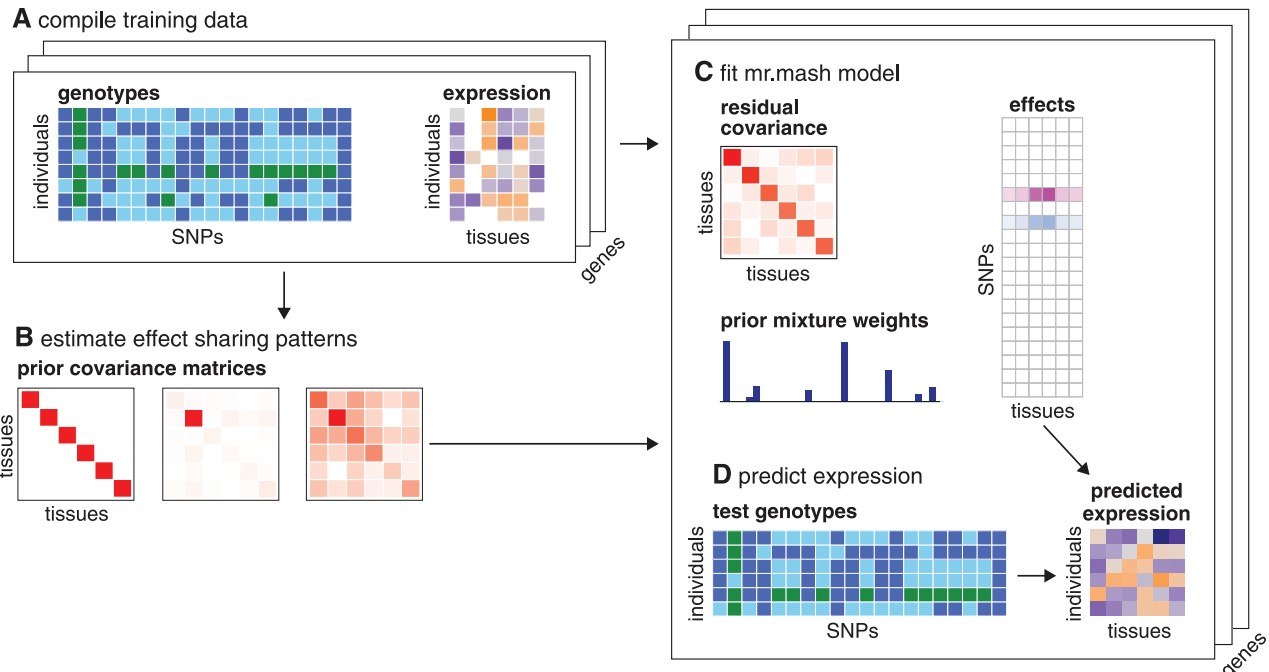

**Fig 1. Overview of mr.mash for predicting multi-tissue gene expression.** The data are the SNP genotypes $X$ and expression levels $Y$ measured in multiple tissues for a selected set of genes (A). *mr.mash* also accepts expression data with missing measurements (depicted as white boxes in A). The *mr.mash* prior (2) may include a mixture of "canonical" covariances (effect sharing patterns) as well as "data-driven" patterns that are learned from the data (B). Once these covariances $S_{0,k}$ are determined, a *mr.mash* model (1–2) is fitted separately for each gene (C). The primary *mr.mash* result is a matrix of coefficients $B$, but fitting a *mr.mash* model also typically involves estimating a residual variance-covariance matrix, $V$, and the weights $w_{0,k}$ controlling the importance of the different covariances $S_{0,k}$ in the prior. The estimated coefficients are often sparse; that is, most of the SNPs have no effect on expression (in C, white boxes depict zeros in $B$). The $B$ estimated by *mr.mash* can then be used to predict gene expression from genotypes (D); see also Eq 3. Note that while this diagram illustrates *mr.mash* for predicting multi-tissue gene expression, this analysis pipeline may be adapted to other settings where multivariate, multiple linear regression is appropriate.

Fig 1 summarizes the workflow for a typical *mr.mash* analysis. A key output of *mr.mash* is the (approximate) posterior mean of the regression coefficients, $\hat{B} := E_q(B)$. This point estimate can be used to predict unobserved outcomes for new samples from their predictor values. Specifically, given predictor values stored as an $n_{\text{new}} \times p$ matrix $X_{\text{new}}$, we can predict the outcomes as

$$\hat{Y}_{\text{new}} := X_{\text{new}}\hat{B}. \tag{3}$$

The variational empirical Bayes approach accomplishes the twin goals of (a) computing posterior effect estimates and (b) adapting the priors to the data while making the underlying computations fast and scalable to large data sets, especially compared with alternative strategies like MCMC [20]. The trade-off is that the approximate posterior distribution obtained with our variational methods will tend to overstate certainty compared with the true posterior distribution [48], and so its use for inference (as opposed to prediction) requires particular care [20]. In this regard, one might consider *mr.mash* more directly comparable to penalized regression methods like the *Elastic Net* [15], which are also more naturally applied to prediction than inference.

## Handling missing data

When analyzing multivariate data, it is common for a large fraction of the $Y$ values to be unavailable, or "missing." For example, in the GTEx expression data [31] (see Applications), the average missing rate is about 60% (after removing a few tissues that are mostly missing). Thus, for broad applicability, it is important for multivariate methods to be able to cope with missing values.

To deal with missing values, we extend the variational approximation to include the posterior distribution of the missing entries (see the S1 Text for details). Computationally, this extension adds a step to the iterative algorithm that "imputes" the missing values. Specifically, denoting $Y_{\mathrm{obs}}$ as the set of observed expression levels and $Y_{\mathrm{miss}}$ as the set of unobserved (missing) expression levels, the approach imputes the missing values $Y_{\mathrm{miss}}$ by computing an approximate posterior distribution for $Y_{\mathrm{miss}}$ given $Y_{\mathrm{obs}}$ and current estimates of the intercept $b_0$, effects $B$, and residual covariance $V$. A similar approach was implemented in [54].

## Software availability

The methods introduced in this paper are implemented as an R package [55] which is available for download at https://github.com/stephenslab/mr.mash.alpha.

# Verification and comparison

## Simulations using GTEx genotypes

We compared *mr.mash* and other methods based on the multivariate, multiple regression model (1), in the task of predicting gene expression in multiple tissues from genetic variant genotypes. To perform systematic evaluations of the methods in realistic settings, we simulated gene expression data for 10 tissues using genotypes from the GTEx project [31]. Specifically, we used the 838 genotype samples generated by whole-genome sequencing. (The GTEx project also collected extensive gene expression data via RNA sequencing, but we did not use these data in our simulations.) The simulated data sets varied considerably in number of genetic variants, from 41 to 21,247 (S1 Text).

We performed simulations under several scenarios; the scenarios differed in the way the effects of the causal variants were simulated. (We use "causal variant" as a shorthand for "genetic variant $j$ having a true non-zero effect in the linear regression for at least one tissue"; that is, $b_j \neq 0$.)

First, we considered three simple simulation scenarios intended to capture "extreme" settings one might encounter in a multivariate analysis:

A. "Equal Effects," in which each causal variant affects all tissues with *the same effect in every tissue*.

B. "Independent Effects," in which each causal variant affects all tissues and *the effects are independent across tissues* (more precisely, the effects are independent conditioned on the genetic variant being a causal variant).

C. "Mostly Null," in which causal variants affect only the first tissue, and therefore the remaining tissues are unaffected by genotype. This represents a scenario in which the genetic effects on gene expression are *tissue-specific*. (To be clear, while the effects of genotype on expression are tissue-specific, in these simulations *the gene is still expressed in all tissues*. For example, this is not the same as a "specifically expressed gene" as defined in [56].)

In all these scenarios, the causal variants explained 20% of the variance of each tissue.

We also considered two more complex scenarios intended to capture a combination of factors that one might encounter in more realistic settings:

D. "Equal Effects + Null," in which the effects on tissues 1 through 3 were equal and explained 20% of the variance of each tissue, and there were no effects in tissues 4 through 10. This represents a scenario where effects are shared only within a subset of tissues.

E. "Shared Effects in Subgroups," in which effects were drawn from a mixture of effect sharing patterns: half of the time, the effects were shared (unequally) across tissues 1 through 3 and explained 20% of the variance of each tissue; otherwise, the effects were shared (unequally) in tissues 4 through 10 and explained only 5% of the variance of each tissue. This scenario was intended to reflect the patterns of effect sharing in the GTEx Project data (see for example Fig. 3a in [35]).

In each Scenario A–E, we simulated 20 gene expression data sets for 20 randomly chosen genes.

Separately for each tissue, we summarized the accuracy of predicted expression levels in test set samples using the commonly used "root mean squared error" (RMSE) metric, defined as

$$\text{RMSE}(s) := \sqrt{\frac{1}{n_{\text{test}}} \sum_{i=1}^{n_{\text{test}}} (y_{is} - \hat{y}_{is})^2}, \tag{4}$$

where $y_{is}$ is the true expression value of tissue $s$ in the $i$th test sample, $\hat{y}_{is}$ is the estimated expression value, and $n_{\text{test}}$ is the number of samples in the test set (which in these experiments was always 168). To make the RMSE more comparable across tissues with different variances we always standardized the RMSE by dividing it by the standard deviation of the true expression measurements in the test set.

See S1 Text for more details about the simulations.

## Methods compared

We compared *mr.mash* with existing multivariate, multiple regression methods: the *Group Lasso* [57] and the *Sparse Multi-task Lasso* [39, 58], both of which use penalties to stabilize and improve accuracy of the fitted models; and a univariate, penalty-based method, the *Elastic Net* [15], applied independently to each tissue. The *Elastic Net* was used in the original PrediXcan method for gene expression prediction in TWAS [11], and therefore we view this approach as a baseline univariate regression method for comparison with the multivariate methods. (We note that recent univariate regression approaches with more flexible priors could yield better predictions in this setting, e.g., [10, 25].) More recently, the *Sparse Multi-task Lasso* was used in UTMOST, a method for cross-tissue expression prediction in TWAS [39]. (To be clear, UTMOST uses the *Sparse Multi-task Lasso*, and not the *Group Lasso*. This was stated incorrectly in [59].) In the results, these three methods are labeled "g-lasso","smt-lasso" and "e-net".

We also assessed the impact of the choice of prior covariance matrices on the performance of *mr.mash*. To do so, we compared three variants of *mr.mash*: (1) *mr.mash* with only "canonical" prior covariance matrices; (2) *mr.mash* with only "data-driven" prior covariance matrices; and (3) *mr.mash* with both types of prior covariance matrices. (See S1 Text for details on these matrices.) We expected that the third variant would adapt well to the widest range of scenarios, and therefore would be the most competitive method overall, with the disadvantage being that it would require more computation. However, we found that *mr.mash* with only data-driven matrices was competitive in terms of prediction accuracy in all the simulated scenarios and

was also faster than the other two variants (S1 Text and S1 and S2 Figs). Therefore, in the comparisons with other methods, we ran *mr.mash* with the data-driven matrices only.

See S1 Text for more details on how the methods were applied to the simulated data sets.

### Results with full data

We begin with the results on the simulations in the "Equal Effects," "Independent Effects" and "Mostly Null" scenarios. Although these scenarios are not the most realistic, they are simpler to understand, and help clarify the behavior of different approaches.

In the Equal Effects scenario, *mr.mash* substantially outperformed the other methods (Fig 2A). In this scenario, the effects of each causal variant were the same in all tissues, and among the methods compared *mr.mash* is unique in its ability to adapt to this scenario; in particular, by adapting the prior to the data, *mr.mash* learned that most of the effects were shared equally or nearly equally across tissues. To illustrate, in one simulation *mr.mash* assigned 81% of the non-null prior weight to matrices capturing equal effects or very similar effects. By contrast, the penalty terms in the penalty-based methods were not flexible enough to adapt to this scenario. Unsurprisingly, the *Elastic Net* performed worst in this scenario because it implicitly

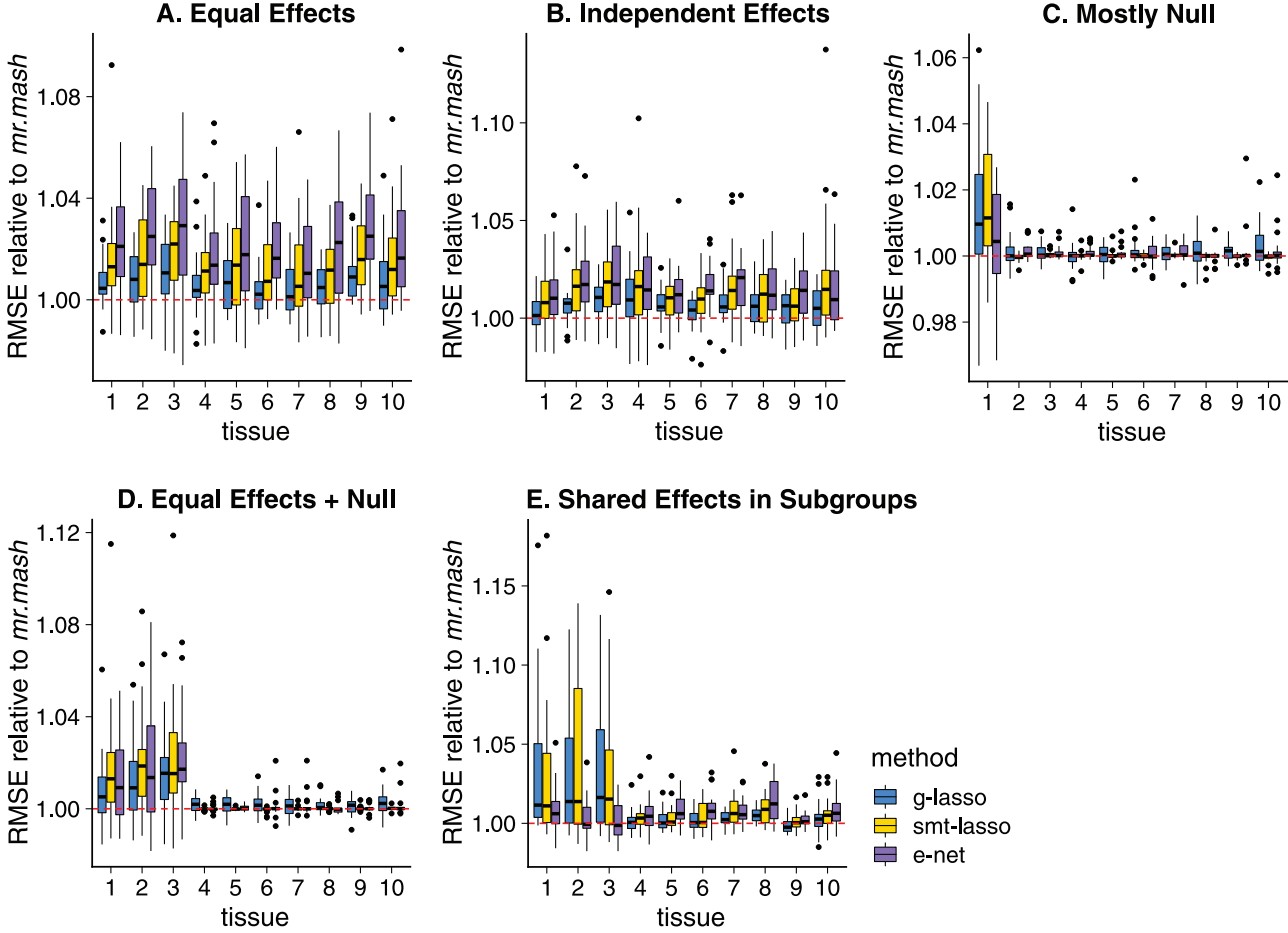

**Fig 2. Prediction accuracy in simulations with full data.** Each plot summarizes the accuracy of the test set predictions in 20 simulations. The thick, black line in each box gives the median RMSE relative to the *mr.mash* RMSE. Since RMSE is a measure of prediction error, lower values indicate better prediction accuracy. Note that the *y*-axis ranges vary among panels.

assumes that the effects are independent, whereas in fact they are highly dependent. Also, *Group Lasso* performed substantially better than the *Sparse Multi-task Lasso* in this scenario; however, this may reflect differences in the way these methods were applied (see S1 Text), rather than a fundamental advantage of the *Group Lasso* over the *Sparse Multi-task Lasso*.

In the Independent Effects scenario (Fig 2B), performance was more similar among the methods. In this scenario there is less to be gained from multivariate regression methods because, once the causal variants are identified, knowing the effect size in one tissue does not help with estimating the effect size in another tissue. Nonetheless, multivariate methods do still have some benefits because they can more accurately identify the casual variants (that is, the variants that have a non-zero effect on at least one tissue). Specifically, the effects for a given genetic variant are either all zero or all non-zero, and all three multivariate methods we consider (*Group Lasso*, *Sparse Multi-task Lasso* and *mr.mash*) can take advantage of this situation. Consequently, the qualitative differences between methods are somewhat similar to the Equal Effects scenario, although the quantitative differences are smaller.

In the Mostly Null scenario (Fig 2C), there is much less benefit to multivariate methods because tissues 2–10 are uncorrelated with the genotypes. In fact, all the methods performed similarly in tissues 2–10. In tissue 1—the one tissue that is partly explained by genotype—the *Group Lasso* and *Sparse Multi-task Lasso* methods performed *worse* than the *Elastic Net*. Consider that the *Group Lasso*'s penalty is poorly suited to the Mostly Null setting—the penalty effectively assumes that effects are either all zero or all non-zero—and because 9 out of the 10 tissues had no genetic effects, the *Group Lasso* penalty strongly encouraged the non-zero effects in tissue 1 toward zero. More surprisingly, the *Sparse Multi-task Lasso* also did not adapt to this scenario, despite having an additional penalty that in principle allows for sparsity across tissues. In contrast to the *Group Lasso* and *Sparse Multi-task Lasso*, *mr.mash*'s prior could adapt to this setting thanks to covariance matrices that allow for tissue-specific effects. Although the prediction accuracy of *mr.mash* in tissue 1 was essentially the same as *Elastic Net*'s, it is nonetheless reassuring that, in contrast to the other multivariate methods, *mr.mash* was no worse than *Elastic Net*.

We now describe the results from the two more complex scenarios, "Equal Effects + Null" and "Shared Effects in Subgroups."

The Equal Effects + Null scenario is a hybrid of the Equal Effects and Mostly Null scenarios, and so the results in Fig 2D reflect those in Panels A and C. As expected, all methods performed similarly in tissues 4–10 (which were uncorrelated with the genotypes), whereas in tissues 1–3 the performance differences were similar to those observed in the Equal Effects scenario, although smaller because here these effects were shared across fewer tissues. As in the Mostly Null scenario, the *Group Lasso* and *Sparse Multi-task Lasso* overshrank the effects in tissues 1–3, whereas *mr.mash* learned to shrink the effects in tissues 1–3 differently from the effects in tissues 4–10, thanks to prior covariance matrices that allowed for strong correlations among tissues 1–3 only. For example, in one simulation *mr.mash* assigned 79% of the non-null prior weight to matrices capturing equal effects or very similar effects in tissues 1–3 and no effects or small effects in the remaining tissues.

The Shared Effects in Subgroups scenario (Fig 2E) is designed to be reflective of actual gene expression studies, and is therefore the most complex of the simulation scenarios we consider. Here all methods performed similarly in tissues 4–10, where the genetic effects explained only a small proportion of phenotypic variance (5%). In tissues 1–3, this scenario includes shared effects (explaining 20% of the phenotypic variance), but the sharing was not quite as strong as in the Equal Effects simulations. As a result, performance gains from conducting a multivariate analysis should be similar to, but not as strong as, the Equal Effects + Null scenario, and the results confirm this. The benefit of *mr.mash* over the *Elastic Net* is more modest in this more

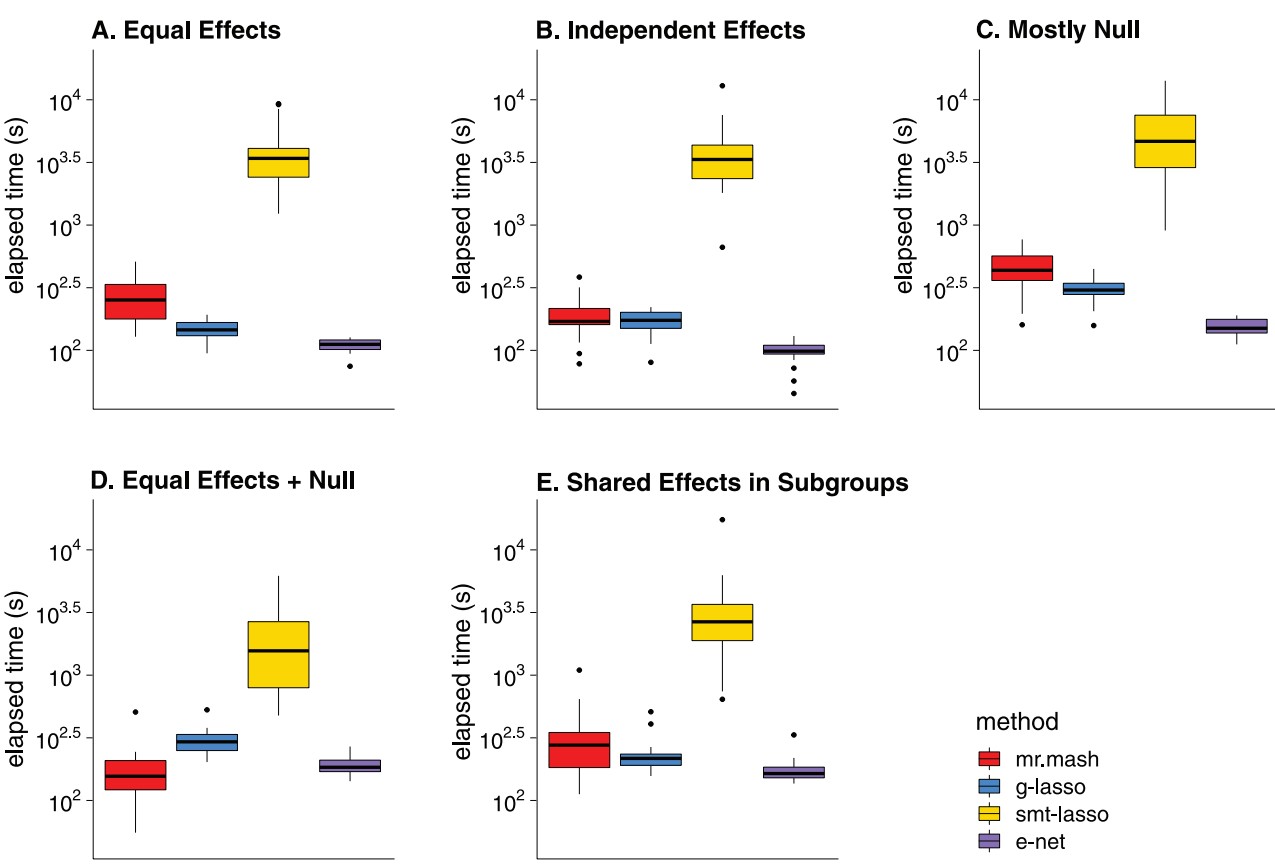

**Fig 3. Runtimes in simulations with full data.** Each plot summarizes the distribution of model fitting runtimes for the 20 simulations in that scenario. The *mr.mash* runtimes do not include the initialization step which was performed using *Group Lasso*. Once model fitting was completed, computing the predictions was very fast for all methods so we did not include the prediction step in these runtimes. See S1 Text for details on the computing environment used to run the simulations. The thick, black line in each box gives the median runtime.

complex scenario, possibly also reflecting the challenge of adapting *mr.mash*'s flexible prior to the complex patterns of effect sharing. Like the Mostly Null and Equal Effects + Null scenarios, the relatively inflexible penalty in the *Group Lasso* cannot capture the complex patterns of sharing, and this explains its inferior performance in tissues 1–3.

We also compared the computational time of the different methods (Fig 3). The runtime of *mr.mash* (with data-driven matrices only) was typically only slightly higher than *Elastic Net* or *Group Lasso*, usually within a factor of 2. Although the *Elastic Net* and *Group Lasso* solved a much simpler optimization problem, they required a more intensive cross-validation step to tune the strength of the penalty term; in contrast, the analogous step in *mr.mash* involved tuning the prior, and was achieved by an empirical Bayes approach that was integrated into the model fitting procedure, thereby reducing the effort of model fitting. The *Sparse Multi-task Lasso* took the longest to run in part because it tuned two parameters by cross-validation, in contrast to the one parameter in the *Elastic Net* and *Group Lasso*. (A more efficient implementation of *Sparse Multi-task Lasso* from [59] performed similarly to the software used in these experiments, but didn't allow for missing data; see S5 and S6 Figs for a comparison of the two *Sparse Multi-task Lasso* implementations, and see S1 Text for details.) A caveat of *mr.mash* is that the dominant computational term scales, at best, quadratically or, at worst, cubically in the number of tissues, $r$ (S1 Text), so for much larger numbers of tissues *mr.mash* may be much slower than the *Elastic Net* or *Group Lasso* which both scale linearly in $r$.

### Results with missing data

We also compared the methods in settings where some measurements were missing. We repeated the simulations as described above, except that we randomly set 70% of the entries of $Y$ to missing before running the methods. For motivation, in the actual GTEx gene expression data about 62% of the entries of $Y$ are missing (they were not measured). Since the R package `glmnet` implementing the *Group Lasso* does not allow for missing values, in these simulations we compared *mr.mash* to the *Elastic Net* and the *Sparse Multi-task Lasso* only. Also, to demonstrate the benefits of integrating data imputation with model fitting, we compared to a naive imputation approach in which the missing values in each column of $Y$ were imputed as the mean for that column, then we ran *mr.mash* with this "mean-imputed" $Y$. This naive approach is labeled "*mr.mash* + mean imputation" in the results.

As in the simulations without missing data, in most of the simulations with missing data *mr.mash* outperformed both the *Elastic Net*, the *Sparse Multi-task Lasso* and *mr.mash* with the naive imputation (Fig 4). Using the *mr.mash* model to impute missing values was most beneficial in situations where the effects were larger or shared more consistently across tissues, which were also the situations without missing data where *mr.mash* was most helpful for improving accuracy.

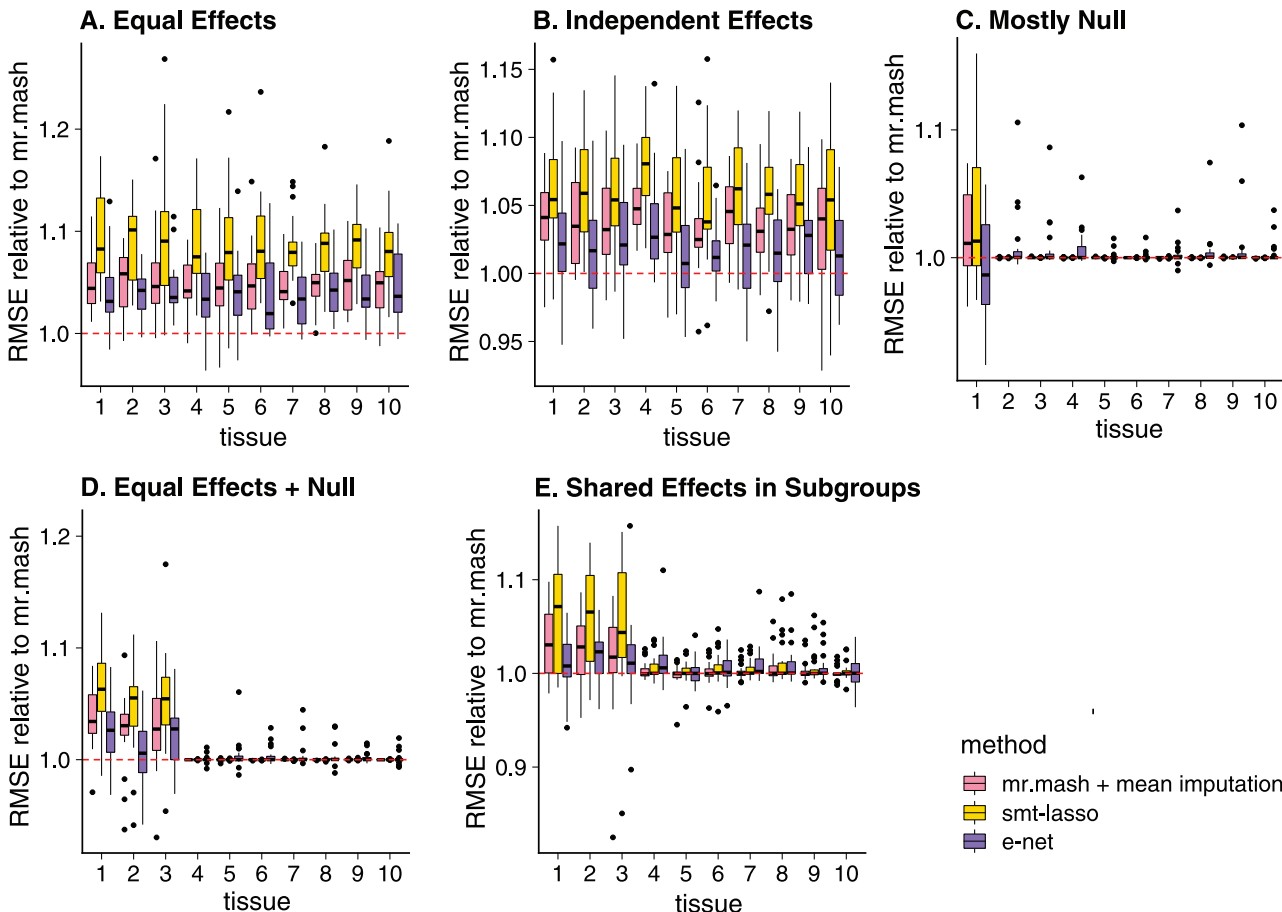

**Fig 4. Prediction accuracy in simulations with missing data.** Each plot summarizes the accuracy of the test set predictions in 20 simulations. The thick, black line in each box gives the median RMSE relative to the *mr.mash* RMSE. Since RMSE is a measure of prediction error, lower values are better. Note that the *y*-axis range varies among panels.

Comparing *mr.mash* to the *Elastic Net* and *Sparse Multi-task Lasso*, the greatest gains in performance were in the Equal Effects and Independent Effects scenarios, and these gains were greater than in the simulations without missing data (compare to Fig 2). We attribute these greater gains to the fact that the effective sample sizes were smaller in these simulations, and therefore there was more potential benefit to estimating effects jointly when the effects were shared across tissues. Only in the Mostly Null scenario did *mr.mash* perform (slightly) worse than *Elastic Net*. This is not unexpected because there was little benefit to analyzing the tissues jointly in this scenario.

We found the *Sparse Multi-task Lasso* performed poorly in all simulations with missing data, even in scenarios such as the Equal Effects and Independent Effects that favor multivariate regression approaches. This was unexpected and suggests that the implementation of this method for missing data may need improvement to be applied in practice.

The introduction of missingness into the simulations increased the differences in computation time; in particular, *Elastic Net* was faster than with full data, whereas *mr.mash* was slower (Fig 5). This was because *Elastic Net* was applied to each tissue separately, and the missing data simply reduced the size of the data sets, whereas *mr.mash* iteratively imputed the missing data, so the expected computational effort was as if *mr.mash* were run on a full data set. *mr.mash* with missing data typically took longer than running *mr.mash* on the mean-imputed data;

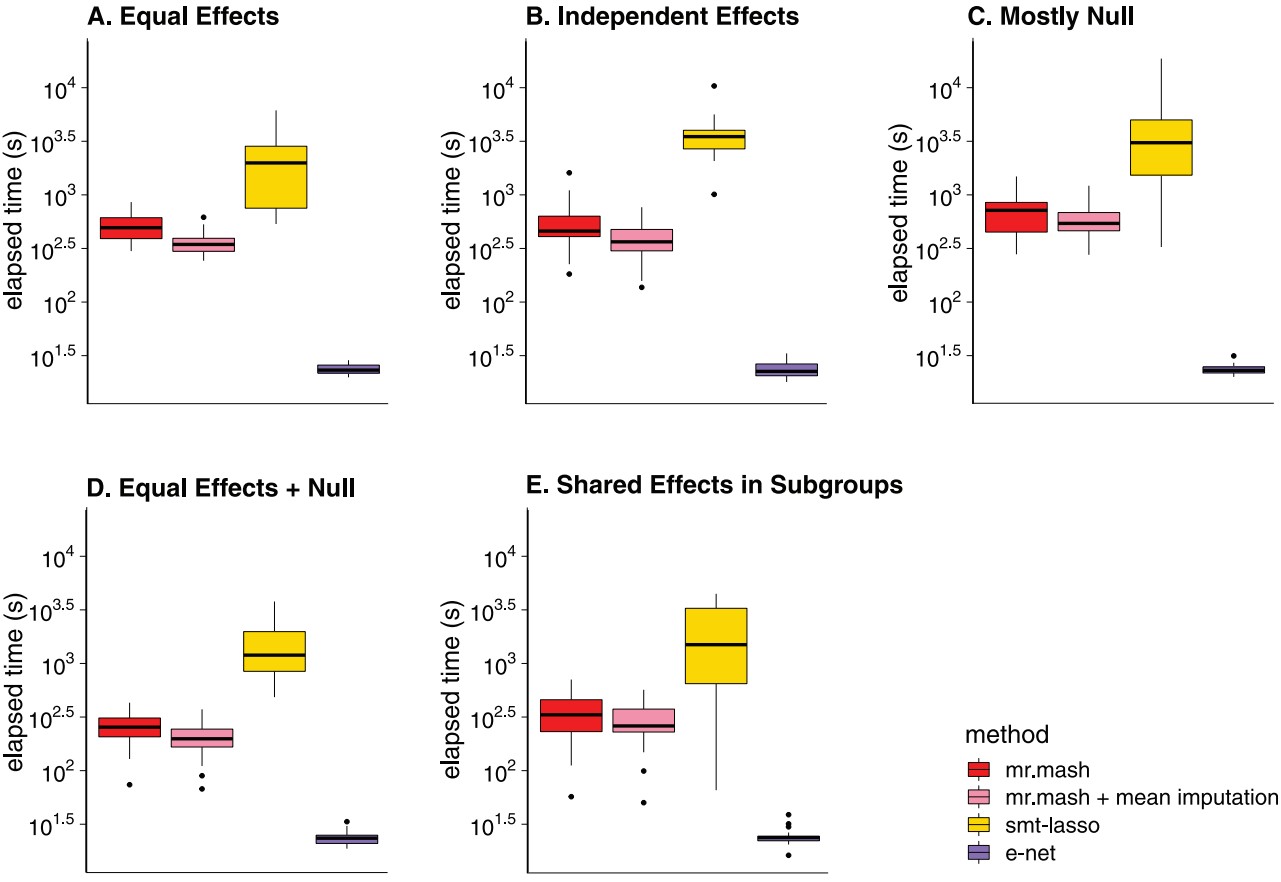

**Fig 5. Runtimes in simulations with missing data.** Each plot summarizes the distribution of model fitting runtimes for the 20 simulations in that scenario. The *mr.mash* runtimes do not include the initialization step which was performed using the *Elastic Net*. Once the model fitting was completed, computing the predictions was very fast for all methods, so we did not include the prediction step in these runtimes. See S1 Text for the details on the computing environment used to run the simulations. The thick, black line in each box gives the median runtime.

indeed, imputing the missing data typically increased the number of iterations needed for the *mr.mash* algorithm to converge to a solution, thereby increasing the overall time involved in model fitting. Like the full-data simulations, the *Sparse Multi-task Lasso* was much slower than the other methods (cautioning again that the software used in these simulations was not as efficient as other available software).

## Applications

### Case study: Predicting gene expression from GTEx data

Finally, we considered an application with real data: using genotypes to predict gene expression in 48 tissues, using data from the GTEx Project. The GTEx data includes *post mortem* gene expression measurements obtained by RNA sequencing and genotypes obtained by whole-genome sequencing for 838 human donors [31]. Since expression measurements were not always available in all 48 tissues, it was important for the multivariate analysis to be able to handle missing data. The tissues varied greatly in the number of available gene expression measurements: among the 48 tissues, skeletal muscle had the most measurements available (706), whereas *substantia nigra* had the least (114) (Fig 6).

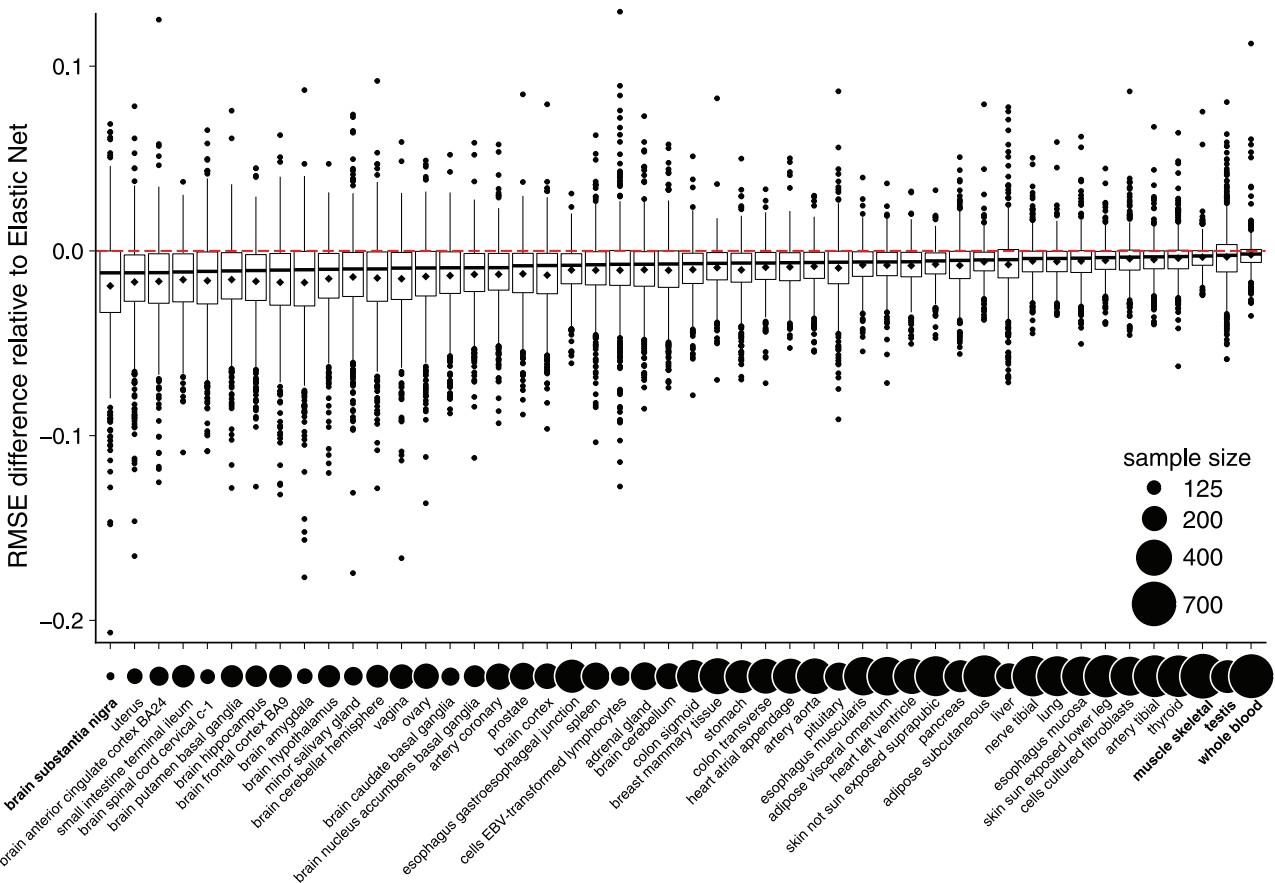

**Fig 6. Accuracy of gene expression predictions in GTEx data.** Relative RMSE differences between the *Elastic Net* predictions and the *mr.mash* predictions in GTEx test samples are plotted along the y-axis as $\frac{\text{RMSE}(mr.mash) - \text{RMSE}(\text{e-net})}{\text{RMSE}(\text{e-net})}$. Each box in the box plot summarizes the relative RMSE differences from predictions for 1,000 genes. Since RMSE is a measure of prediction error, lower values are better. Below the boxes in the box plot, the circles are linearly scaled in area by the number of available gene expression measurements in each tissue. Tissues mentioned in the text are highlighted in bold.

Using these data, we compared *mr.mash* and the *Elastic Net* for predicting expression from unseen (test) genotypes. (We also performed a more limited comparison with the *Sparse Multi-task Lasso*; see below.) We analyzed 1,000 genes chosen at random, and for each gene we used all genetic variants within 1 Mb of the gene's transcription start site (also removing genetic variants not satisfying certain criteria for inclusion; see S1 Text). To assess the prediction accuracy of each method, we randomly split the 838 GTEx samples into 5 subsets and performed 5-fold cross-validation; that is, we fit the model using a training set composed of 4 out of 5 subsets, then we assessed prediction accuracy in the fifth subset. We repeated this 5 times for each of the 5 splits and summarized prediction accuracy as the average RMSE in the 5 test sets. Prediction accuracy varied considerably with gene and tissue because some genes in some tissues were more strongly predicted by genetic variant genotypes. Therefore, to make results more comparable across genes, we reported *relative* performance accuracy—specifically, the relative difference in RMSE between the two methods, using the *Elastic Net* as a reference point, $\frac{\text{RMSE}(mr.mash) - \text{RMSE}(\text{e-net})}{\text{RMSE}(\text{e-net})}$.

These comparisons are summarized in Fig 6. Overall, *mr.mash* produced substantially more accurate gene expression predictions, although the improvement varied considerably from gene to gene and from tissue to tissue. Anecdotally, the improvements tended to be greatest for tissues with more sharing of effects and/or for tissues with smaller sample sizes (Fig 6 and S3 Fig). In such tissues, the improvement in accuracy was more reflective of the Equal Effects or Independent Effects simulations. For example, the *substantia nigra* brain tissue had the fewest measurements and benefited from strong sharing of effects with other brain tissues. This strong sharing among the brain tissues is illustrated by the top covariance matrix in the *mr.mash* prior (Fig 7).

In contrast, tissues with the largest sample sizes and more tissue-specific eQTLs tended to show less improvement with multivariate analysis. For example, testis, whole blood and skeletal muscle had weaker sharing of effects (Fig 7), consistent with earlier analyses [31, 35]. In such tissues, there was still some benefit to *mr.mash*, but the gains were more reflective of the Shared Effects in Subgroups or Mostly Null simulations.

We also compared *mr.mash* to the *Sparse Multi-task Lasso*. However, due to the very long running time of the *Sparse Multi-task Lasso* software in these data sets, we performed a more limited comparison on only 10 randomly chosen genes. We fit the *Sparse Multi-task Lasso* to the training set, increasing the size of the grid of the two penalty parameters to 50 in an attempt to improve its performance (in the simulations we used a smaller grid of 10 points to reduce computation). The results of this comparison (S4 Fig) illustrate the tendency of the *Sparse Multi-task Lasso* to overshrink effect size estimates, to the point that, in many cases, the scaled RMSE was 1, implying that all the estimated coefficients were exactly zero. *mr.mash* achieved a lower RMSE than the *Sparse Multi-task Lasso* in most cases.

## Discussion

We have introduced *mr.mash*, a Bayesian multiple regression framework for modeling multiple (e.g., several dozen) responses jointly, with accurate prediction being the main goal. A key feature of our approach is that it can learn patterns of effect sharing across responses from the data, then use the learned patterns to improve prediction accuracy. This feature makes our method flexible and adaptive, which are advantages of particular importance for analyzing large, complex data sets. Our method is also fast and computationally scalable thanks to the use of variational inference (rather than MCMC) for model fitting.

Although we focussed on a specific application—predicting gene expression from genotypes—*mr.mash* is a general method that could be applied to any problem calling for

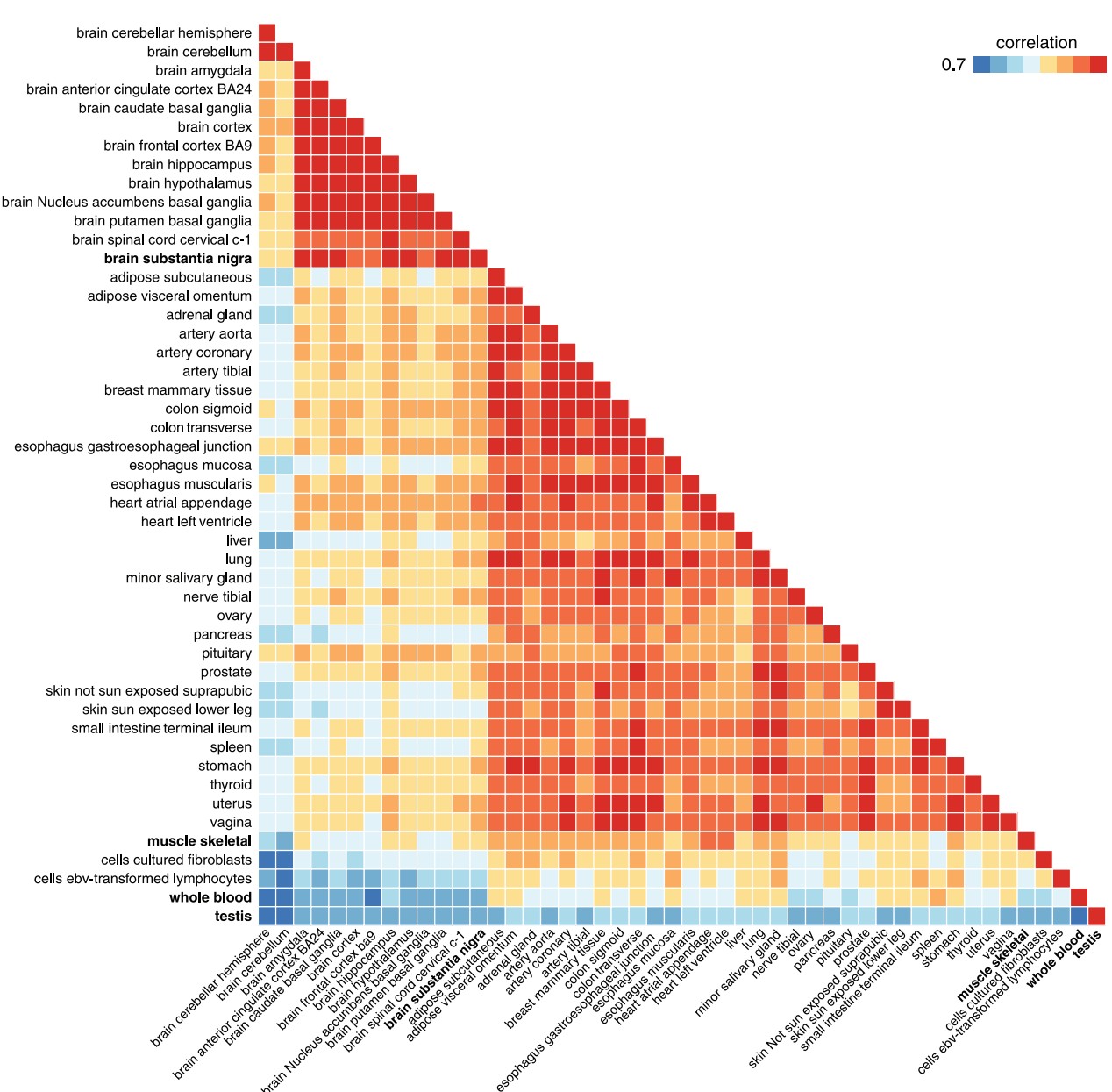

**Fig 7. Top pattern of eQTL sharing in GTEx data identified by *mr.mash*.** This heatmap shows the prior covariance matrix $S_{0,k}$ that had the largest total weight in the prior (that is, the total prior weight across the 1,000 genes). This covariance matrix was scaled to obtain the correlation matrix shown above. Tissues mentioned in the text are highlighted in bold.

multivariate, multiple regression. This includes, for example, breeding value prediction for multiple related phenotypes in agricultural settings and polygenic score computation for multiple populations in human genetics. Indeed, recent work, performed independently but using a similar approach, showed improved accuracy in cross-ancestry prediction [26]. In these applications, the number of causal variants is typically much larger than for gene expression phenotypes, which could lead to larger improvements in prediction accuracy. While we expect *mr.mash* to be slower in such whole-genome regression applications, it is scalable in that its computational complexity (per iteration) is linear in the number of samples and in the number of predictors (genetic variants).

To demonstrate that *mr.mash* can indeed reasonably scale to whole-genome data sets, we ran *mr.mash* on a data set with 10 phenotypes, 4,901 individuals and 441,627 SNPs. (The phenotypes were simulated so that 1,000 randomly selected SNPs explained 50% of variance in each phenotype. The genotypes were from the type 1 diabetes case-control cohort from [60].) With $K = 250$ mixture components in the prior, *mr.mash* took about 50 hours to converge to a solution within the chosen tolerance (a change in ELBO less than 0.01) using 4 CPUs on a machine equipped with dual-core Intel Xeon Gold 6348 CPUs. On a machine with Apple M1 Ultra CPUs, the model fitting algorithm took roughly 24 hours to converge. Clearly, applying *mr.mash* to much larger multi-trait data sets, and in particular for data sets with hundreds of thousands of individuals and millions of genetic variants ("biobank-scale" data sets), will require some additional innovation. One possible approach would be to adapt *mr.mash* to work with "summary data" [25, 61, 62].

A limitation of *mr.mash* is that it is not ideally suited for selecting among highly correlated variables (which has, for example, been the emphasis of statistical fine-mapping methods [1, 2, 24, 62, 63]). This is because the variational approximation used in *mr.mash* cannot capture the strong dependence in the posterior distribution for the effects of highly correlated variables. Indeed, if two variables are perfectly correlated, and one is causal, *mr.mash* will select one at random and exclude the other [20]. (This behavior is also displayed by the Lasso [41].) Therefore, in settings where variable selection is the main goal, alternative approaches (e.g., [24]) may be preferred. On the other hand, since selecting randomly among correlated variables does not diminish prediction accuracy [20], *mr.mash* can perform well for prediction problems even when highly correlated variables are present.

## Supporting information

**S1 Fig. Prediction accuracy of *mr.mash* variants in simulations with full data.** Each plot summarizes the accuracy of the test set predictions in the 20 simulations for that scenario. The three methods compared were: (1) *mr.mash* with only "canonical" prior covariance matrices; (2) *mr.mash* with only "data-driven" prior covariance matrices; and (3) *mr.mash* with both types of prior covariance matrices. The thick, black line in each box gives the median RMSE relative to the "data-driven" *mr.mash* RMSE. Since RMSE is a measure of prediction error, lower values are better. Note that the *y*-axis range varies among panels.
(PDF)

**S2 Fig. Runtimes for *mr.mash* variants in simulations with full data.** Each plot summarizes the distribution of model-fitting runtimes for the 20 simulations in that scenario. Note the runtimes did not include the initialization step, which was implemented by running the *Group Lasso* on the same data set. Once the model fitting was completed, computing the predictions was very fast, so we did not include the prediction step in these runtimes. See S1 Text for the details on the computing environment used to run the simulations. Note that the *y*-axis range varies among panels.
(PDF)

**S3 Fig. Relationship between improvement in prediction accuracy and GTEx tissue sample size.** Tissues are plotted along the x-axis by the number of available gene expression measurements and along the y-axis by the improvement in RMSE relative to the *Elastic Net*; that is, (RMSE(*mr.mash*) − RMSE(e-net))/RMSE(e-net).
(PDF)

**S4 Fig. Comparison of *mr.mash* vs. *Sparse Multi-task Lasso* for 10 randomly chosen genes in GTEx data.** Each plot compares the accuracy of the *mr.mash* and *Sparse Multi-task Lasso*

gene expression predictions in test samples for a single gene, separately for each tissue. The prediction accuracy is summarized as the RMSE relative to the RMSE that would be obtained by the "naive" predictor in which the genotype has no effect on expression (the naive predictor is therefore simply the mean of the expression measurements in the training data); that is, the x-axis shows RMSE(smt-lasso)/RMSE(naive) and the y-axis shows RMSE(mr.mash)/RMSE (naive). Note that some genes are not expressed in all tissues and so some plots have fewer than 48 points.
(PDF)

**S5 Fig. Prediction performance comparison of *Sparse Multi-task Lasso* implementations in simulations with full data.** Each plot summarizes the accuracy of the test set predictions in 20 simulations for that scenario. Accuracy was quantified by the (standardized) RMSE so that lower RMSE means better accuracy. The two implementations compared are the `mtlasso` Python software (https://github.com/aksarkar/mtlasso) and the R and C++ implementation used in [59] (this was labeled `multi_tissue_twas_sim` in the figure because it was downloaded from a git repository with this name, https://github.com/RitchieLab/multi_tissue_twas_sim). Note that the data sets used in this comparison were not the same as the ones used in the main full-data simulations; for this comparison, the data sets were simulated the exact same way except that synthetic genotypes were used instead of the genotypes from the GTEx Project. For more details on this comparison, see [64], in particular the file `mrmash_vs_mtlasso_vs_utmost.html`.
(PDF)

**S6 Fig. Runtimes comparison of *Sparse Multi-task Lasso* implementations in simulations with full data.** Each plot summarizes the distribution of model-fitting runtimes for the 20 simulations in that scenario. For details on the methods compared, see the caption for S5 Fig. See also S1 Text for the details on the computing environment used to run the simulations.
(PDF)

**S1 Text. Detailed methods.** Detailed description of the methods, including: preparation of GTEx data; simulations with GTEx genotypes; methods compared in the simulations; derivations of *mr.mash* algorithms with full data; and derivations of *mr.mash* algorithms with missing data.
(PDF)

## Acknowledgments

We thank the University of Chicago Research Computing Center for providing high-performance computing resources used to run the numerical experiments. We thank Jeff Spence and Jonathan Pritchard for helpful discussions.

## Author Contributions

**Conceptualization:** Fabio Morgante, Matthew Stephens.

**Data curation:** Fabio Morgante, Gao Wang, Yuxin Zou.

**Formal analysis:** Fabio Morgante, Gao Wang.

**Funding acquisition:** Fabio Morgante, Matthew Stephens.

**Investigation:** Fabio Morgante, Peter Carbonetto.

**Methodology:** Fabio Morgante, Peter Carbonetto, Yuxin Zou, Abhishek Sarkar, Matthew Stephens.

**Project administration:** Fabio Morgante, Matthew Stephens.

**Resources:** Fabio Morgante, Peter Carbonetto, Gao Wang, Matthew Stephens.

**Software:** Fabio Morgante, Peter Carbonetto, Gao Wang, Yuxin Zou, Abhishek Sarkar.

**Supervision:** Matthew Stephens.

**Validation:** Fabio Morgante, Peter Carbonetto, Matthew Stephens.

**Visualization:** Fabio Morgante, Peter Carbonetto.

**Writing – original draft:** Fabio Morgante, Peter Carbonetto, Gao Wang, Abhishek Sarkar, Matthew Stephens.

**Writing – review & editing:** Fabio Morgante, Peter Carbonetto, Gao Wang, Yuxin Zou, Abhishek Sarkar, Matthew Stephens.

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
