## [Decision Letter · Decision Letter 0]

23 Jan 2023

Dear Dr Morgante,

Thank you very much for submitting your Methods entitled 'A flexible empirical Bayes approach to multivariate multiple regression, and its improved accuracy in predicting multi-tissue gene expression from genotypes' to PLOS Genetics.

The manuscript was fully evaluated at the editorial level and by independent peer reviewers. The reviewers appreciated the attention to an important problem, but raised some substantial concerns about the current manuscript. Based on the reviews, we will not be able to accept this version of the manuscript, but we would be willing to review a much-revised version. We cannot, of course, promise publication at that time.

If you decide to revise the manuscript for further consideration at PLOS Genetics, please aim to resubmit within the next 60 days, unless it will take extra time to address the concerns of the reviewers, in which case we would appreciate an expected resubmission date by email to plosgenetics@plos.org.

We are sorry that we cannot be more positive about your manuscript at this stage. Please do not hesitate to contact us if you have any concerns or questions.

Yours sincerely,

Xiaofeng Zhu

Section Editor

PLOS Genetics

Xiaofeng Zhu

Section Editor

PLOS Genetics

Reviewer's Responses to Questions

**Comments to the Authors:**

Reviewer #1: Morgante et al present mr mash, a method for creating genetic prediction models. The key advantage of mr mash is that it is multi-trait, and can share genetic architecture across traits. They first show that mr mash outperforms three rival multi-trait prediction tools on simulated data, then provide comparable results on real data.

Overall, I consider the paper well written, and the proposed model novel. There is also a novel way of handling missing phenotypic values. My main limitation is that mr mash is only demonstrated on small gene expression sized data - 600 or so individuals for 10-50 traits, and using what seems to be only a small number of SNPs (I cant find the number mentioned). And therefore, I have no confidence the method can be used on genome-wide data for large numbers of individuals (e.g., 10,000s of individuals, 100,000s SNPs). While I do not consider it a requirement that the data can be applied to large numbers of individuals / SNPs, failure to do so greatly limits the utility of the method. My second limitation, is that the authors do not appear to compare mr mash against state-of-the-art uni-trait methods (I elaborate on this below). On a simliar note, my third limitation is that the authors do not appear to compare their new method of imputing phenotypes to the alternative of mean-imputation.

Major comments

1 - I approve of the use of variational bayes (VB), as I found it very effective for my 2021 prediction method (Nature Communications). In particular, by VB version of BayesR was many times (50+) more efficient than the MCMC version of BayesR (note that my use of VB was inspired by Bolt-LMM, however, I subsequently saw some of the authors of this paper had championed VB before Bolt-LMM, so I apologise for not being aware of those works earlier). HOWEVER, I experimented with parameter selection based on the approximate likelihood (which I believe is what you do), and found it performed poorly compared to cross-validation. I guess this is also what Bolt-LMM authors found, as they also recommend CV. It may be that for your applications (much smaller), ELBO is fine. However, please can you evaluate the effectiveness of parameter selection by ELBO and CV

2 - In the discussion, you suggest mr mash would scale to large sized data, this seems very speculative for me. As discussed above, the number of individuals and SNPs might both be 1000 times larger, and thus even with linear scaling, 2*9 seconds scaled up is very very large. Further, as you state, the architecure would be very different to gene expression data (i.e., 1-10 causal variants explaining 20% heritability would almost certainly not be realistic), and so this might affect performance / convergence.

3 - I believe you should compare mr mash to a leading uni-trait method (instead of only to three other multi-trait methods). As well as being good practice, I note you claim a use of mr mash is for improving the accuracy / power of TWAS, and so a uni-trait analysis would indicate whether this is likely to be the case (because currently, TWAS are based on results from uni-trait prediction models).

In terms of which uni-trait method to compare with, personally, I found that LDAK-Bolt-Predict was the best performing uni-trait method, outperforming rival uni-trait methods such as BayesR, Bolt-LMM, BLUP, Lasso, SBLUP, SBAYESR, LDpred, Lassosum, AnnoPred, LDpred-funct and PRS-CS. While you should always be wary of authors finding their own method performs best, I note that Dr Oliver Pain independently found the same (https://opain.github.io/GenoPred/). The advantage of LDAK-Bolt-Predict came from incorporating alternative heritability models (in particular, most methods assume the "GCTA Model", whereas I used the "BLD-LDAK Model"). By contrast, it seems mr mash assumes the "van Raden / alpha=0" model (because you centre, but do not standardize SNPs), which I have found better than GCTA Model but inferior to LDAK-Thin or BLD-LDAK Model. My guess is that the choice of heritability models has limited impact for your analyses, because you consider very few SNPs. Further, the mr mash model appears to have a good prior distribution choice. For these reasons, it seems likely mr mash will outperform a leading uni-trait method, but it remains good to confirm.

4 - Please can you compare your clever phenotype imputation method to the naive approach of mean-imputation.

Minor coments.

5 - I liked the range of simulations, and agreed with your view they covered a wide range of plausible extremes.

6 - I could not find the number of SNPs mentioned (nor an approximate number). I see in the real data, you restricted to within 1Mb of each gene. This suggests you are analysing about 5Mb regions, which with sequenced (common?) variants, means about 15k variants in each 5Mb region (but of course, I should not have to guess!)

7 - Runtime - I find reporting time as log_base 2 seconds quite hard to interpret! Maybe minutes would be more clear?

8 - Figures - I appreciate you mention y-axes can vary, this is very upfront of you. However, given they do not vary much (imo), would it be better to fix them to the same values?

9 - I am unsure what the order of tissues is in Figure 5 - but it seems median RMSE difference. Would it make more sense to instead rank by number of individuals (I personally prefer agnostic ordering, rather than ordering based on results).

10 - It seems a bit weird to focus on the whole-sequencing data in GTEX - gives the impression you want to restrict the number of individuals, or want very dense data.

11 - Page 6 - if you report standardized RMSE, why not just report correlation? Those two might be equivalent (but latter is perhaps easier to explain and more intuitive)

12 - Page 6 - I am generally satisfied with your choice of comparison methods, except the sentence "For the Elastic Net, the mixing parameter α was set to 0.5". I am not familiar with your references, but this seems a strange choice. E.g., when I used glmnet, I thought I was advised to use a range of values (e.g., something like 0.5, 0.1 and 0.01), and I am feel this is normally what I see in applications.

13 - I appreciated your sensitivity analysis regarding prior covariance matrix

14 - Some typos

Page 5 - "the cuasal variants"

Page 6 - "Shared Sffects in Subgroups"

#Signed Doug Speed

Ps, I was forced to do this review a bit faster than normal (sorry), so I apologise if I missed a few key details (e.g., asked you to do analyses you had done already).

Reviewer #2: The authors proposed a Bayesian multivariate mixture regression model to jointly predict multiple phenotypes using genotype data (e.g., gene expression in 48 tissues in GTEx). The main contribution is the coordinate descent algorithms via the variational inference and the use of data-driven prior to capture covariance between the target tasks (e.g., sharing effect among GTEx tissues). Overall, the paper is well written and easy to follow. The mr.mash method was also described in great amount of details in the supplementary text. The experiments using simulation and GTEx data were also well justified. I have several major comments or suggestions for the authors to consider.

Major comments:

Comments on the Supplementary text (S1_text):

1. In S1_text, is MAF of 0.05 based on GTEx or 1000 Genome?

2. Algorithm 1, for line 10, 15, and 16, please append the equation numbers, which I believe should be Eq 11, 47, and 12, respectively.

3. Also, I felt that using the subscript of “1” in Algorithm 1 is confusing where subscript “j” should have been used for each variant j

4. For Eq 11, please write down the closed-form update equation.

5. E.4.3. The authors mentioned that they used flashr to generate the data-driven prior covariance and the number of covariance matrices produced differ depending on the simulation setting. For Equal effects and equal effects + Null, the pipeline somehow generates 9 matrices. What do those 9 matrices represent?

6. Related to point 5, when applied to the GTEx data (when the groundtruth effects are not known), they said each run produced between 33 and 35 data-driven covariance matrices. What are those? Does each covariance correspond to a distinct component so in other words you ended up 33-35 mixture components? I think this is one of the most important contributes and need elaboration.

7. What’s “Extreme Deconvolution” algorithm?

Main text:

Methods:

1. The description of the method is too brief. The technical contributions of the method was not well highlighted in my opinion. I suggest write down the key EM variational closed-form updates in Method description section of the main text. In particular, in E-step, consider adding closed-form update for the mean and covariance of the effect size b_j (Eq 20, 21) but changed it to the actual update (not the simple regression) for b_j_bar and S_bar; in M-step: the key update step for V (Eq 12) and mixing proportion. The derivation can stay in the supplement but the method should be self-contained in the main text for reproducibility.

2. In the main text, it would be also helpful to describe the intuition of the closed-form updates. For example, the mean-field update of the mean effect size of SNP j takes into account the correlation between SNP j and the residual error R_j when regressing out the effects of all SNPs except SNP j as well as the covariance between traits.

3. Does mr.mash work with summary statistics? I think it can at least when assuming the data are completely observed. That will also improve scalability of the method.

Simulation Results:

1. Although authors briefly discussed this, how does mr.mash perform differently on sparse and dense LD in simulation and/or real data?

2. On line 232, in Independent Effects scenario, authors wrote “multivariate methods do still have some benefits because they can more accurately identify the causal variants”. But as we know and as authors mentioned as well, accurate prediction does not imply accurate fine-mapping. Therefore, I don’t think this point is a valid in justifying why mr.mash performs better in the Independent Effects scenario than the univariate methods.

3. Fig. 2. Please change seconds to minutes or hours for the ease of reading the runtime.

4. As scalability is a main highlight of the method, what’s the computing time as a function of the number of SNPs, number of tasks, and sample size?

5. In line 289, authors wrote “mr.mash scales either quadratically or cubically in r”. When is it quadratic and when is it cubic? I thought that it is always cubic because of inverting the covariance matrix V.

6. What’s the exact time complexity of mr.mash?

GTEx application:

1. Authors experimented on 1000 randomly chosen genes. It would be useful to examine three more choices:

a. (1) tissue-specific genes (i.e., similar to the mostly null effect scenario);

b. (2) highly variable genes (i.e., similar to the shared effects in subgroups scenario);

c. (3) most heritable genes across tissues (i.e., examining the upper bound of prediction performance)

2. Line 354, authors wrote “the improvements tended to be greatest for tissues with more sharing effects and for tissues with smaller sample sizes (Fig 5, S3 Fig)”. Fig 5 only supports the latter claim on the sample size but not the form claim on the sharing effects. Similar to S 3, I suggest plot delta RMSE as a function of average covariance computed from Fig 6 to see whether there is a linear trend.

3. For Fig. 6, please label substantia nigra to help the reader find it when being referred to this figure in the main text.

4. Line 372: authors commented on the tendency of smt-LASSO to overshrink effect size. By looking at Fig. S4, smt-lasso is not much different from taking the mean of the gene expression, implying that only the unpenalized bias term in the linear model was fit and all regression coefficients are essentially zeros. Would this be caused by large penalty lambda weights, which are supposed to be selected via a validation set?

5. As authors cited, Hu et al (2019) described a method called UTMOST,which was applied to GTEx. While authors experimented their own implementation of smt-lasso, it is worthwhile to directly experiment UTMOST (https://github.com/Joker-Jerome/UTMOST).

Minor:

1. Typo in line 171: “Sffects”

2. Line 314: “may need improving” => “may need improvement”

**Have all data underlying the figures and results presented in the manuscript been provided?**

Reviewer #1: Yes

Reviewer #2: None

PLOS authors have the option to publish the peer review history of their article (what does this mean?). If published, this will include your full peer review and any attached files.

Reviewer #1: **Yes: **Doug Speed

Reviewer #2: No

---

## [Decision Letter · Decision Letter 1]

13 Apr 2023

Dear Dr Morgante,

Thank you very much for submitting your Methods entitled 'A flexible empirical Bayes approach to multivariate multiple regression, and its improved accuracy in predicting multi-tissue gene expression from genotypes' to PLOS Genetics.

The manuscript was fully evaluated at the editorial level and by independent peer reviewers. The reviewers appreciated the attention to an important problem, but raised some substantial concerns about the current manuscript. Based on the reviews, we will not be able to accept this version of the manuscript, but we would be willing to review a much-revised version. We cannot, of course, promise publication at that time.

Both reviewers have a concern regarding the format of response letter. Therefore, we ask you to provide a point- by-point response letter.

If you decide to revise the manuscript for further consideration at PLOS Genetics, please aim to resubmit within the next 60 days, unless it will take extra time to address the concerns of the reviewers, in which case we would appreciate an expected resubmission date by email to plosgenetics@plos.org.

We are sorry that we cannot be more positive about your manuscript at this stage. Please do not hesitate to contact us if you have any concerns or questions.

Yours sincerely,

Xiaofeng Zhu

Section Editor

PLOS Genetics

Xiaofeng Zhu

Section Editor

PLOS Genetics

Reviewer's Responses to Questions

**Comments to the Authors:**

Reviewer #1: In general, the authors have responded to all my comments (the exception is their response to my minor comment 6, as explained below, but this is a very minor point so I do not feel this requires "correcting")

Here are some responses:

Major comment 1 - thanks for explaining the CV issue, that makes sense

Major comment 3 - sorry, it seems I did miss that comparison

Minor comment 6 - Sorry if my comment was confusing. I often use genotyped snps to mean "snps on a standard genotyping chip", whether obtained through direct genotyping, imputation or sequencing (I think others do the same, but I may be wrong). So while you only have sequence data, you could do a test where you reduced this to just (eg) 600k "genotyped" snps. However, as I say, this was a minor point, so I require no further analysis.

Otherwise, thanks for your comprehensive replies

Lastly, I did not like your response format :) I think it is more normal to reply in line with my comments. In particular, I first read your response on my telephone, and it was too fiddly to try and scroll up and down from my comments back to yours to check you had answered them all!

Reviewer #2: I'm not familiar with the style the response letter was written. The authors should have provided specific response right after *each* comment rather than enlisting all my comments and then writing a long statement that responds to only some of my comments. This is not really a "point-by-point" response and makes it hard to find where each of my comments was addressed.

While I can see and appreciate that some of my comments were carefully considered, some of them were missed because of the format of the response letter. Therefore, I’m sorry but I would like to hold on and kindly suggest the authors to redo the response letter in a style similar to the following point-by-point format (Thank you in advance for your patience):

Comments on the Supplementary text (S1_text):

1. In S1_text, is MAF of 0.05 based on GTEx or 1000 Genome?

Authors’ response:

2. Algorithm 1, for line 10, 15, and 16, please append the equation numbers, which I believe should be Eq 11, 47, and 12, respectively.

Authors’ response:

3. Also, I felt that using the subscript of “1” in Algorithm 1 is confusing where subscript “j” should have been used for each variant j

Authors’ response:

**Have all data underlying the figures and results presented in the manuscript been provided?**

Reviewer #1: Yes

Reviewer #2: None

PLOS authors have the option to publish the peer review history of their article (what does this mean?). If published, this will include your full peer review and any attached files.

Reviewer #1: **Yes: **Doug Speed

Reviewer #2: No

---

## [Decision Letter · Decision Letter 2]

4 May 2023

Dear Dr Morgante,

Thank you very much for submitting your Methods entitled 'A flexible empirical Bayes approach to multivariate multiple regression, and its improved accuracy in predicting multi-tissue gene expression from genotypes' to PLOS Genetics.

The manuscript was fully evaluated at the editorial level and by independent peer reviewers. The reviewers appreciated the attention to an important topic but identified some concerns that we ask you address in a revised manuscript.

Reviewer 2 still has some concerns. We therefore ask you to modify the manuscript according to the reviewer 2's recommendations. Your revisions should address the specific points made by the reviewer.

Yours sincerely,

Xiaofeng Zhu

Section Editor

PLOS Genetics

David Balding

Section Editor

PLOS Genetics

Reviewer's Responses to Questions

**Comments to the Authors:**

Reviewer #2: 1. I would not recommend use Pseudocode as Figure 1. But I do think that the Peudocode is helpful and can be moved to the Method section. Perhaps some figurative illustration like a graphical abstract would help highlight the input and output of the algorithm.

2. In my comment on the GTEx application: I asked the following question on real data not simulated data and would like to see some efforts in addressing them experimentally. This is not circular if only the training data are used. Alternatively, if all the ~20,000 genes are experimented, then we can examine the estimated effect sizes for each gene and group the genes based on scenarios as outlined in the simulation to gain novel insight on the gene regulatory programs. For example, do the tissue-specific estimated effect sizes indeed capture known tissue-specific genes?

3. Related to point 2, how robust is the heatmap in Fig. 7. In other words, if you fit your method on another randomly sampled 1000 genes, would the heatmap change?

“Authors experimented on 1000 randomly chosen genes. It would be useful to examine three more choices:

1) tissue-specific genes (i.e., similar to the mostly null effect scenario); For this, authors can use the same way as used in Finucane et al (2018) (Nature Genetics 50, 1-14 (2018)) to define tissue-specific genes.

2) highly variable genes (i.e., similar to the shared effects in subgroups scenario);

3) most heritable genes across tissues (i.e., examining the upper bound of prediction performance)”

Other comments:

I added some clarification to my following comments. Overall, the method description should be self-contained or else leave out the unnecessary details.

1. I wrote “E.4.3. The authors mentioned that they used flashr to generate the data-driven prior covariance and the number of covariance matrices produced differ depending on the simulation setting. For Equal effects and equal effects + Null, the pipeline somehow generates 9 matrices. What do those 9 matrices represent? “ referring to this sentence “The number of data-driven matrices produced by this pipeline was different for each simulation, although in all the Equal Effects and Equal Effects + Null simulations this pipeline always produced 9 matrices.” The 9 matrices should be described in the text as it adds confusion to why there are 9 not say 5 or 11 matrices as it current reads. Also, the github repo referred by the authors has no minimal documentation explaining this point ().

2. In one of my comments, I wrote “Related to point 5, when applied to the GTEx data (when the groundtruth effects are not known), they said each run produced between 33 and 35 data-driven covariance matrices. What are those? Does each covariance correspond to a distinct component so in other words you ended up 33-35 mixture components? I think this is one of the most important contributes and need elaboration.” Authors responded that the ideas are borrowed from Urbut et al (2019). Sorry for being meticulous on this but that does not really answer my question. Please add more explanation.

3. “What’s “Extreme Deconvolution” algorithm?” Authors cited Extreme Deconvolution algorithm by (Bovy et al., 2011). Could you further describe the rationale on why the Extreme Deconvolution is used and briefly its core ideas.

4. The github repo has no minimal documentation on how to run the method. Tutorial and vignette is needed.

Minor comments:

Authors mentioned “quadratically” but I think it should be just "cubically" since all of the time complexity functions involve cubic of r

**Have all data underlying the figures and results presented in the manuscript been provided?**

Reviewer #2: Yes

PLOS authors have the option to publish the peer review history of their article (what does this mean?). If published, this will include your full peer review and any attached files.

Reviewer #2: No

---

## [Editor Report · Decision Letter 3]

2 Jun 2023

Dear Dr Morgante,

We are pleased to inform you that your manuscript entitled "A flexible empirical Bayes approach to multivariate multiple regression, and its improved accuracy in predicting multi-tissue gene expression from genotypes" has been editorially accepted for publication in PLOS Genetics. Congratulations!

Yours sincerely,

Xiaofeng Zhu

Section Editor

PLOS Genetics

Xiaofeng Zhu

Section Editor

PLOS Genetics

Comments from the reviewers (if applicable):

**Data Deposition**

http://datadryad.org/submit?journalID=pgenetics&manu=PGENETICS-D-22-01332R3

**Press Queries**

---

## [Editor Report · Acceptance letter]

5 Jul 2023

PGENETICS-D-22-01332R3 

A flexible empirical Bayes approach to multivariate multiple regression, and its improved accuracy in predicting multi-tissue gene expression from genotypes 

Dear Dr Morgante, 

We are pleased to inform you that your manuscript entitled "A flexible empirical Bayes approach to multivariate multiple regression, and its improved accuracy in predicting multi-tissue gene expression from genotypes" has been formally accepted for publication in PLOS Genetics! Your manuscript is now with our production department and you will be notified of the publication date in due course.

With kind regards,

Lilla Horvath

PLOS Genetics

On behalf of:
